# Multiepitope fusion protein-based ELISA for enhanced brucellosis serodiagnosis

Tiansong Zhan[1☯], Yan Li[2,3,4☯], Yujia Xie[1☯], Shuangshuang Li[5], Lili Lian[6*], Wei Han[7*], Dehui Yin[1*]

**1** Jiangsu Engineering Research Center of Biological Data Mining and Healthcare Transformation, Xuzhou Medical University, Xuzhou, Jiangsu, China, **2** Shandong Center for Disease Control and Prevention, Jinan, China, **3** Shandong Provincial Key Laboratory of Intelligent Monitoring, Early Warning, and Prevention and control for Infectious Diseases, Jinan, China, **4** Shandong Institute of Preventive Medicine, Jinan, China, **5** Department of Oncology, Heze Municipal Hospital, Heze, China, **6** Department of Laboratory Medicine, First Hospital of Jilin University, Changchun, China, **7** Department of Laboratory Medicine, the Second Affiliated Hospital of Xuzhou Medical University, Xuzhou Mining Group Genera Hospital, Xuzhou, China

☯ These authors contributed equally to this work
* lianlili@jlu.edu.cn (LL); xkzyyjyk@163.com (WH); yindh16@xzhmu.edu.cn (DY)

## Abstract

### Background

Brucellosis is a widespread zoonotic disease with approximately 2.1 million new human cases annually. Traditional serological methods, which rely on lipopolysaccharide (LPS) as a diagnostic antigen, suffer from cross-reactivity with other Gram-negative bacteria. To address these issues, we aimed to develop a multiepitope fusion protein for improved brucellosis diagnosis.

### Methods

We identified linear B-cell epitopes from *Brucella* proteins using the Immune Epitope Database (IEDB) and constructed a multiepitope fusion protein. The fusion protein was expressed through prokaryotic expression, purified, and was evaluated using an indirect enzyme-linked immunosorbent assay (iELISA) with serum samples from patients with confirmed brucellosis (n = 279) and negative controls from febrile patients (n = 126). To rigorously assess cross-reactivity, it was also tested against a separate panel of sera from 283 non-brucellosis febrile patients with laboratory-confirmed infections by other bacterial pathogens.

### Results

The engineered fusion protein, consisting of 11 optimized linear B-cell epitopes derived from the consolidation of 23 initial epitopes obtained from the IEDB, was assessed using iELISA. This evaluation yielded an area under the curve (AUC) of 0.9912 relative to LPS, demonstrating a sensitivity of 95.34% and a specificity of

**Data availability statement:** All data generated or analysed during this study are included in this published article and its supplementary information files.

**Funding:** This work was supported by QingLan Project of Jiangsu Province Grant to DY (2024) and Postgraduate Research & Practice Innovation Program of Jiangsu Province Grant to YX (Grant number KYCX25-3209). The funders had no role in study design, data collection and analysis, decision to publish, or preparation of the manuscript.

**Competing interests:** The authors have declared that no competing interests exist.

93.65% in comparison to the negative control group (n = 126). Critically, when tested against the distinct cross-reactivity panel, the fusion protein exhibited cross-reactivity with 9 serum samples from 283 patients infected with other bacterial pathogens, whereas LPS showed cross-reactivity in 41 out of 283 samples.

## Conclusions

When evaluated against a clinically relevant control cohort of non-brucellosis febrile patients, the fusion protein demonstrated a substantial advantage by exhibiting significantly lower cross-reactivity compared to LPS, which frequently cross-reacted with other bacterial infections. The multiepitope fusion protein developed in this study demonstrates significant potential as a diagnostic tool for brucellosis. However, the study's limitations, including a small sample size and lack of information on *Brucella* species, suggest the need for further research with larger and more diverse sample sets to fully validate its clinical application.

### Author summary

This study developed a multiepitope fusion protein for improved diagnosis of human brucellosis. Linear B-cell epitopes for *Brucella* proteins were identified from the Immune Epitope Database (IEDB) and constructed a fusion protein comprising 11 optimized epitopes (derived from 23 initial epitopes). The fusion protein was expressed, purified, and evaluated using indirect enzyme-linked immunosorbent assay (iELISA) with serum samples from confirmed brucellosis and non-brucellosis patients. Results showed the iELISA using the fusion protein had an area under the curve (AUC) of 0.9912, with 95.34% sensitivity and 93.65% specificity, outperforming lipopolysaccharide (LPS) in diagnostic performance and cross-reactivity. The fusion protein exhibited low cross-reactivity with bacteremia infections (9 out of 283 sera), while LPS showed substantially higher cross-reactivity (41 out of 283 samples). Although the study demonstrated the fusion protein's potential as a diagnostic tool for brucellosis, its limitations, including a small sample size and lack of *Brucella* spp. information, suggest the need for further research to fully validate its clinical application.

## 1. Introduction

Brucellosis is a zoonotic infection of global significance, with an estimated 2.1 million new cases of human brucellosis reported annually [1]. The lack of specific clinical symptoms associated with brucellosis complicates early diagnosis, often resulting in misdiagnosis and the potential for chronic infection, which can have severe detrimental consequences [2,3]. Therefore, the accurate diagnosis of *Brucella* infection is critically important [3]. Currently, the diagnosis of brucellosis predominantly relies on

serological methods, including tube agglutination tests, the tiger-red plate test, and enzyme-linked immunosorbent assays (ELISA) [4,5]. Traditional serological methods utilize lipopolysaccharide (LPS) as a diagnostic antigen; however, LPS is also found in other Gram-negative bacteria, leading to cross-reactivity with other bacteria such as *Yersinia enterocolitica* O9 and *Escherichia coli* O157:H7 during the serological diagnosis of brucellosis [6,7]. Consequently, current research in the serological diagnosis of brucellosis concentrates on the identification of appropriate alternative antigens.

Numerous studies in recent research have focused on the development of multiepitope fusion proteins through the application of bioinformatics techniques [8,9]. This approach entails predicting epitopes from *Brucella* antigenic proteins and assembling them into multiepitope fusion proteins for use as diagnostic antigens following prokaryotic expression [10,11]. In this methodology, the selection of *Brucella* antigenic proteins is therefore of paramount importance. A significant body of research has concentrated on *Brucella* outer membrane proteins, among other candidates; nonetheless, no antigenic protein has been identified that rivals LPS in efficacy [12]. Although bioinformatics technology has generated valuable insights for the identification of alternative antigens, it is important to acknowledge its limitations. Specifically, the predicted epitopes primarily consist of linear epitopes that can be expressed in prokaryotic systems, while the conformational epitopes, which represent the majority of epitopes, remain unexpressed [13,14]. Furthermore, the predictions made by bioinformatics methods are often associated with a significant degree of uncertainty. For instance, the prediction accuracy rates of Bepipred and SVMTriP are reported to be 65.93% and 55.2%, respectively [15,16]. In contrast, traditional methods for epitope characterization and identification involve the validation of overlapping synthetic peptides on an individual basis, which incurs substantially higher costs compared to bioinformatics predictions [17,18].

The Immune Epitope Database (IEDB, www.iedb.org) is a publicly accessible resource that is funded by the National Institute of Allergy and Infectious Diseases (NIAID). This database compiles experimental data on antibody and T cell epitopes that have been studied in humans and various animal species, with a particular focus on infectious diseases, allergies, autoimmunity, and transplantation [19,20]. Our search of the IEDB revealed a significant number of experimentally validated linear B-cell epitopes associated with *Brucella*. As a result, we aggregated the documented *Brucella* linear B-cell epitopes from the IEDB, constructed a multi-epitope fusion protein, and evaluated its potential utility for the diagnosis of brucellosis.

## 2. Methods

### 2.1. Ethics statement

All methods were carried out in accordance with the Declaration of Helsinki. The study was reviewed and approved by the Ethics Committee of Xuzhou Medical University (approved number: xzhmu-2024Z052) and verbal informed consent was obtained from all subjects.

### 2.2. Serum samples

A total of 100 positive serum samples and 96 negative serum samples were obtained from the Xuzhou Center for Disease Control and Prevention, while 179 positive serum samples and 30 negative serum samples were sourced from the Shandong Center for Disease Control and Prevention. Each positive sample was validated as positive through the Standard Agglutination Test (SAT), while all negative serum samples were obtained from febrile patients suspected of brucellosis but subsequently confirmed to be negative by SAT. All negative and positive samples were collected from the same respective regions within Xuzhou and Shandong. Moreover, to rigorously evaluate cross-reactivity, serum samples from 283 non-brucellosis febrile patients with laboratory-confirmed infections of other pathogens were obtained from the First Affiliated Hospital of Jilin University. These included patients with confirmed bacterial infections (e.g., *Escherichia coli*, *Klebsiella pneumoniae*, *Pseudomonas aeruginosa*, *Staphylococcus aureus*) and other common pathogens (as detailed in S2 File). The diagnoses were confirmed by microbial culture, PCR, or specific serological tests as part of the hospital's routine diagnostic procedures. Importantly, all these patients were confirmed to be negative for brucellosis by the SAT to exclude any co-infection or misdiagnosis.

## 2.3. Selection of *Brucella* Linear B-cell Epitopes

The identification of linear B-cell epitopes for *Brucella* was performed utilizing the IEDB (https://www.iedb.org/) with the following fundamental parameters [21]: Epitope type: Linear peptide; Assay type: B Cell; Outcome: Positive; Epitope Source: *Brucella*. All other parameters were maintained at their default settings. The retrieved B-cell epitopes were subsequently optimized for integration. In instances where amino acid sequences overlapped, the epitope was regenerated to ensure that the integrated epitope encompassed the overlapping amino acid sequence. After that, BLAST analysis was performed to verify the specificity of the selected *Brucella* linear B-cell epitopes.

## 2.4. Construction and evaluation of *Brucella* multiepitope fusion protein

The optimized integrated *Brucella* linear B-cell epitopes were tandemly linked, with adjacent epitopes connected by the linker sequence 'GGGS' to construct their corresponding amino acid sequences. Subsequently, the aliphatic index, instability index, isoelectric point (pI) and molecular weight (MW) of the protein were predicted for the amino acid sequence of the tandemly linked fusion protein using ProtParam (http://web.expasy.org/protparam/), which is available on the ExPASy website. The physicochemical properties of the fusion proteins were thoroughly evaluated. Additionally, the three-dimensional molecular model of the fusion protein was predicted using I-TASSER (http://zhanglab.ccmb.med.umich.edu/I-TASSER/) to assess the spatial conformation of the fusion protein. The antigenicity of the fusion proteins was predicted and evaluated using VaxiJen software (http://www.ddg-pharmfac.net/vaxijen/VaxiJen/VaxiJen.html, threshold: 0.4, default).

## 2.5. Preparation and purification of fusion proteins

In the preparation of fusion proteins, established references were employed to guide the prokaryotic expression and purification processes [10,11]. The specific methodology is outlined as follows:

**2.5.1. Plasmid construction.** The amino acid sequence of the designed multiepitope fusion protein was reverse-translated and codon-optimized for expression in *Escherichia coli* (*E. coli*). A 6×His tag was incorporated at the C-terminus to facilitate purification. The optimized gene sequence was synthesized de novo and cloned into the pET-30a(+) expression vector between the Nde I and Xho I restriction sites.

**2.5.2. Transformation.** The constructed expression plasmid was transformed into *E. coli* BL21 (DE3) competent cells using a standard heat-shock method. Briefly, the plasmid was mixed with competent cells on ice, heat-shocked at 42°C for 90 seconds, and then placed back on ice. After the addition of LB medium and a recovery incubation, the cells were plated on LB agar plates containing the appropriate antibiotic and incubated overnight at 37°C.

**2.5.3. Protein expression.** For initial small-scale expression, a single transformed colony was inoculated into 1.5 mL of LB medium and cultured at 37°C with shaking at 200 rpm until the $OD_{600}$ reached 0.6-0.8. Protein expression was induced by adding isopropyl β-D-1-thiogalactopyranoside (IPTG) to a final concentration of 0.5 mM, followed by incubation at 37°C for 2 hours. The bacterial cells were then harvested by centrifugation, resuspended in 10 mM Tris-HCl (pH 8.0), and lysed by boiling in 2×SDS loading buffer. Expression of the target protein was analyzed by 12% SDS-PAGE.

For large-scale protein production, the above culture was scaled up to 250 mL of LB medium. When the $OD_{600}$ reached 0.6-0.8, IPTG was added to a final concentration of 0.5 mM, and induction was continued at 37°C for 4 hours. The bacterial cells were harvested by centrifugation at 8000 rpm for 6 minutes. The cell pellet was resuspended in 20–30 mL of 10 mM Tris-HCl (pH 8.0) and lysed by ultrasonication (500 W, 180 cycles of 5 s sonication with 5 s intervals). The soluble fraction (supernatant) and insoluble fraction (precipitate) were separated by centrifugation at 12,000 rpm for 10 minutes and analyzed by 12% SDS-PAGE.

**2.5.4. Protein purification.** The nickel affinity column (Ni Sepharose 6 Fast Flow, GE Healthcare) was initially washed with deionized water to achieve a pH of 7.0. Subsequently, the nickel was adjusted to a pH range of 2–3. The column was then washed again with deionized water to restore a pH of 7.0. Following this, the nickel column was equilibrated with

approximately 100 mL of 10 mM Tris-HCl buffer at pH 8.0, followed by an additional equilibration with approximately 50 mL of the same buffer supplemented with 0.5 M NaCl. The samples were diluted appropriately for analysis, ensuring that the final concentration of sodium chloride in the samples was 0.5 M. At the conclusion of the sample run, the column was washed with 10 mM Tris-HCl buffer (pH 8.0) containing 0.5 M sodium chloride. Protein elution was performed using 10 mM Tris-HCl buffer (pH 8.0) containing varying concentrations of imidazole: 15 mM, 60 mM, and 300 mM, each supplemented with 0.5 M NaCl. The effectiveness of the protein purification process was assessed using 12% SDS-PAGE electrophoresis, and protein quantification was conducted utilizing the BCA Protein Quantification Kit (Beyotime, Shanghai, China). The purity of the purified fusion protein was determined by analyzing the Coomassie Brilliant Blue-stained SDS-PAGE gel using Image Lab software (Bio-Rad, USA). The software calculates the percentage purity based on the integrated intensity of the target protein band relative to the total integrated intensity of all protein bands in the lane. The purified protein was subjected to endotoxin removal and subsequent quantification utilizing the Protein Endotoxin Removal Kit (C0268S, Beyotime, China) and the Endotoxin Detection Kit employing the Limulus Reagent Dynamic Turbidimetric Method (C0271S, Beyotime, China).

## 2.6. Development of the IELISA method

The purified fusion protein was utilized, and a standardized protocol was employed to develop the ELISA method [22].

The fusion protein was diluted to a concentration of 5 µg/mL using a carbonate buffer solution (CBS, pH 9.6) and subsequently added to a 96-well microplate (Corning, USA) at a volume of 100 µL per well. The microplate was incubated overnight at 4°C, followed by three washes with PBST, utilizing 300 µL per well for each wash. A blocking solution composed of 5% skimmed milk powder was then added at a volume of 300 µL per well, and the plate was incubated at 37°C for 2 hours. After an additional round of washing with PBST, serum samples were introduced at a dilution of 1:200 in PBS, with 100 µL added to each well, and incubated at 37°C for 1 hour. Following further washes with PBST, 100 µL of HRP-conjugated rabbit anti-human IgG (diluted 1:10,000, Thermo Fisher, USA) was added to each well and incubated at 37°C for 1 hour. The plate was washed three times with PBST, after which TMB substrate solution was added, and the plate was incubated in the dark for 10 minutes to allow for color development. The reaction was terminated by the addition of 2 M $H_2SO_4$, and the $OD_{450}$ was measured using a microplate reader (Versa Max microplate reader, MD, USA). Laboratory-stored LPS (provided by the China Animal Health and Epidemiology Center, 3 mg/mL, extracted and purified from *B. abortus* A19) was utilized as a control antigen [23]. Serum samples were tested in triplicate following the same procedure. The sensitivity, specificity, area under the curve (AUC), and cutoff values were determined through receiver operating characteristic (ROC) curve analysis.

## 2.7. Evaluation of cross-reactivity via Indirect ELISA Method

The cross-reactivity of the fusion protein and LPS was assessed by testing them against the panel of sera from SAT-negative, non-brucellosis febrile patients (n = 283) described in section 2.1. A sample was considered cross-reactive if its OD450 value exceeded the cut-off value established by the ROC curve.

## 2.8. Statistical methods

Dot plot and ROC curve analyses were conducted using GraphPad Prism version 6.05. Statistical analyses were performed using unpaired Student's *t*-test, with a significance level set at $P < 0.05$.

## 3. Results

### 3.1. Selection of B-cell epitopes

A comprehensive search conducted within the IEDB yielded a total of 23 epitopes associated with the following proteins: Cu-Zn superoxide dismutase (SOD), 31 kDa outer-membrane immunogenic protein precursor (Omp31), chaperone protein DnaK, Omp2b, 26 kDa periplasmic immunogenic protein precursor (BP26), and elongation factor Tu (EF-Tu) (S1 File).

Through a process of merging overlapping sequences, these 23 initial epitopes were consolidated into 11 non-redundant, optimized epitopes (Table 1). Following the BLAST analysis, the selected epitopes, which exhibit a high degree of conservation in classical *Brucella* species, were found to be well conserved among the primary pathogenic species of *B. melitensis* and *B. abortus*. The percent identity observed is 100% (Table 1 and S4 File).

### 3.2. Construction and evaluation of the fusion protein

The fusion protein comprises a total of 342 amino acids. Using the ProtParam tool available on the ExPASy server, we calculated the aliphatic index and isoelectric point of the fusion protein. The aliphatic index is 67.02, and the instability index is computed to be 33.46, this classifies the fusion protein as stable (< 40 indicates stability). The grand average of hydropathicity (GRAVY) being -0.460 indicates that the protein may exhibit relatively good hydrophilic properties. The molecular weight of the constructed fusion protein was predicted to be approximately 35.1 kDa, and its theoretical isoelectric point (pI) was determined to be 4.57. The overall prediction for the protective antigen was calculated to be 0.9559, indicating that it is a probable antigen. The 3D structure of the protein is illustrated in Fig 1B.

### 3.3. Preparation of fusion protein

The multiepitope fusion protein was successfully expressed in *E. coli* BL21(DE3). Solubility analysis revealed that the majority of the recombinant protein was present in the soluble fraction of the cell lysate (Fig 2), which facilitated subsequent purification. The protein was efficiently purified using Ni-NTA affinity chromatography under native conditions. As shown in Fig 2C, SDS-PAGE analysis of the purified product displayed a single predominant band at approximately 45 kDa. Analysis of the purified protein using Image Lab confirmed that the final purity exceeded 90%. After endotoxin removal, the endotoxin level of fusion protein was 0.084 EU/mL.

**Table 1. The 11 optimized B-cell epitopes for fusion protein, derived from the integration of 23 initial epitopes retrieved from IEDB.**

| Protein Source | Position in fusion protein | Epitope | Starting Position | Ending Position | Specificity* |
|---|---|---|---|---|---|
| BP26 | 1 | TMLAAAPDNSVPIAAGENSYNVSVNVVFEIK | 220 | 250 | *B. melitensis*, *B. abortus*, *B. ovis* |
| BP26 | 2 | KKAGIEDRDLQTGGINIQPIYVYPDDKNNLKEPTITGY | 87 | 117 | *B. melitensis*, *B. abortus*, *B. suis*, *B. ovis* |
| DnaK | 3 | SSKDDVVDADYEEIDDNKKSS | 617 | 637 | *B. melitensis*, *B. abortus*, *B. suis*, *B. canis*, *B. neotomae* |
| EF-Tu | 4 | QTREHIL | 110 | 116 | *B. melitensis*, *B. abortus*, *B. suis* |
| Omp2b | 5 | VIEEWAAKVRGDVNITDQFSVWLQGAYSSAATPDQNYGQWG | 255 | 295 | *B. melitensis*, *B. abortus*, *B. suis*, *B. canis*, *B. ovis* |
| Omp31 | 6 | SWTGGYIGINAGYAGGKFKHPFSSFDKEDNEQVSGSLDVTAGGFV | 39 | 83 | *B. melitensis*, *B. suis* |
| Omp31 | 7 | QAGYNWQLDNGVVLGA | 87 | 102 | *B. melitensis*, *B. abortus*, *B. suis*, *B. canis*, *B. ovis* |
| Omp31 | 8 | MVYGTGGLAYGKVKSAFNLGDDAPALHTWSDKTKAGWTLGAGAE | 149 | 192 | *B. ovis* |
| Omp31 | 9 | EYLYTDLGKRNLVDVDNSFL | 204 | 223 | *B. melitensis*, *B. abortus*, *B. suis* |
| Cu-Zn SOD | 10 | APGEKDGKIVPA | 75 | 86 | *B. melitensis*, *B. abortus*, *B. neotomae* |
| Cu-Zn SOD | 11 | LAEIKQRSLMVHVGGDNYSDKPEPLGG | 136 | 162 | *B. melitensis*, *B. abortus*, *B. neotomae* |

*Percent Identity is 100.0% in classical *Brucella* spp. by BLAST analysis.

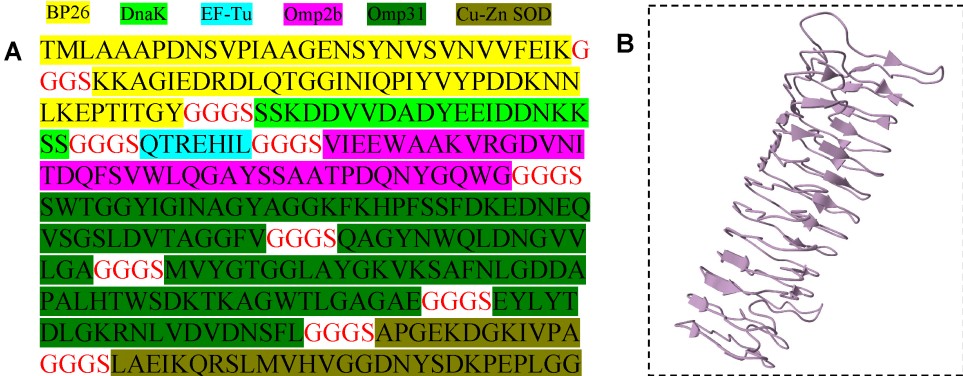

**Fig 1. Construction and evaluation of fusion protein.** (A) Amino acid sequence of the fusion protein; (B) 3D structural models of fusion proteins predicted by I-TASSER.

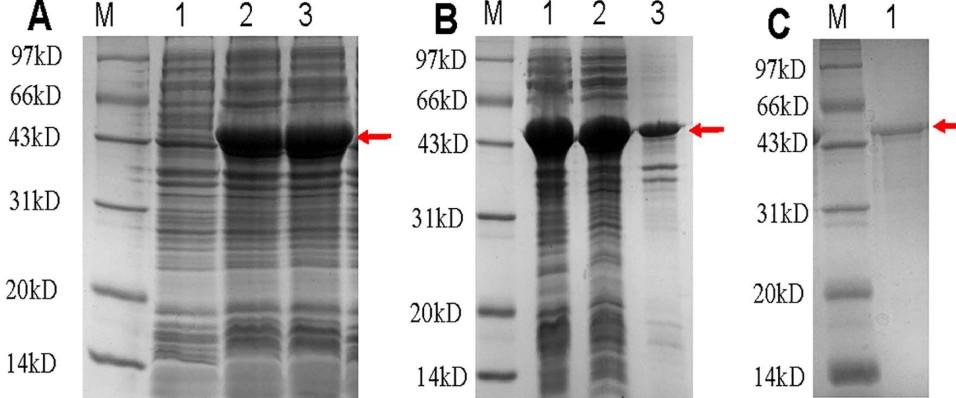

**Fig 2. 12% SDS-PAGE results.** (A) Protein expression results in small amounts. M, Marker; Lane 1, uninduced control (BL21); Lane 2-3, IPTG induced (BL21). (B) Protein bulk expression results. M, Marker; Lane 1, whole bacteria after sonication; Lane 2, supernatant after sonication; Lane3, precipitate after sonication. (C) Protein purification results. M: Marker; Lane 1, purified protein.

### 3.4. Results of iELISA

Following the analysis of the ROC curve, the AUC for the fusion protein in serum was determined as the Youden index. The reactivity profiles of positive and negative serum samples with both antigens are presented as a dot plot in Fig 3A, showing clear separation between the groups. The fusion protein demonstrated excellent diagnostic accuracy against the negative control samples (n = 126), with the AUC of 0.9912 (95% CI: 0.9857 - 0.9968). At the optimal cut-off value ($OD_{450}$ > 0.4861), the assay achieved a sensitivity of 95.34% and a specificity of 93.65%. For comparison, the LPS-based iELISA tested against the same negative control samples showed an AUC of 0.9958, with a sensitivity of 96.77% and a specificity of 98.41% at its cut-off ($OD_{450}$ > 0.3185). The comprehensive evaluation metrics, including accuracy, positive predictive value (PPV), and negative predictive value (NPV), are summarized in Table 2, and raw data is presented in S2 File.

However, when we integrate the control samples (409 sera include 126 negative serum and 283 cross-reactive samples), For, the AUC of Fusion Protein is 0.9930, with a sensitivity of 99.64% and a specificity of 92.18%. For LPS, the AUC is 0.9807, with a sensitivity of 93.91% and a specificity of 94.38% (Table A and Fig D in S3 File).

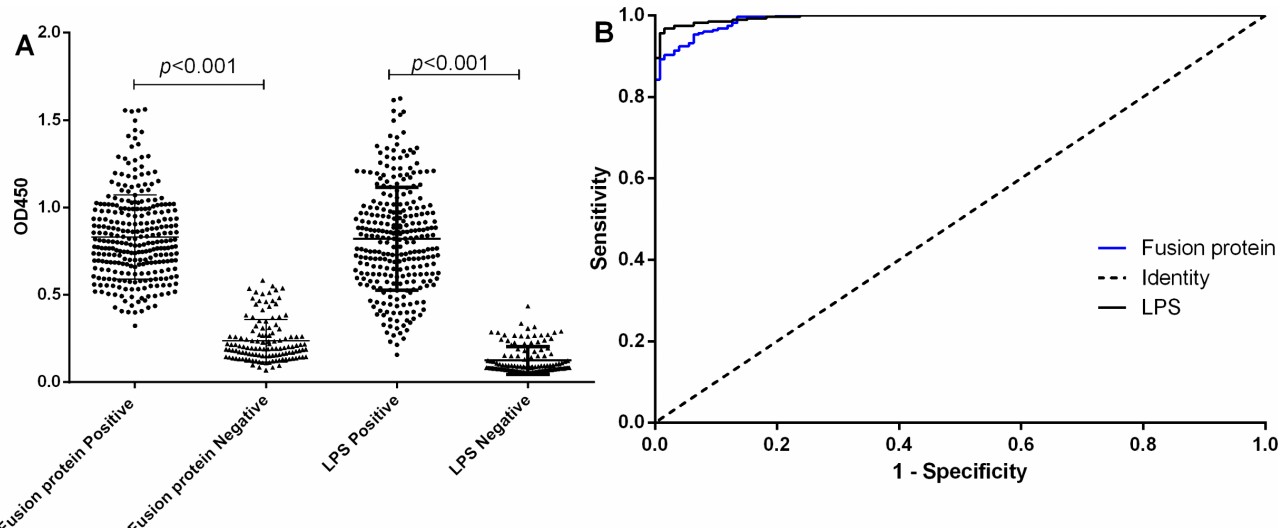

**Fig 3. I-ELISA analysis of human serum samples.** (A) Dot plot of human serum samples. Fusion Protein Positive: OD450 values of positive sera tested with the fusion protein. Fusion Protein Negative: OD450 values of negative sera from the negative control samples (n=126) tested with the fusion protein. LPS Positive: OD450 values of positive sera tested with LPS. LPS Negative: OD450 values of negative sera from the negative control samples (n=126) tested with LPS. (B) ROC analysis of human sera.

**Table 2. Evaluation of iELISA results of the recombinant proteins against the positive and 126 negative control samples.**

| Antigen | AUC | Cut-off value | Sensitivity (95%CI) | Specificity (95%CI) | Positive | | Negative | | Accuracy (%) | PPV (%) | NPV (%) |
|---|---|---|---|---|---|---|---|---|---|---|---|
| | | | | | TP | FN | TN | FP | | | |
| Fusion Protein | 0.9912 (0.9857 to 0.9968) | >0.4861 | 0.9534 (0.9216 to 0.9750) | 0.9365 (0.8787 to 0.9722) | 266 | 13 | 118 | 8 | 94.81 | 97.08 | 90.08 |
| LPS | 0.9958 (0.9924 to 0.9993) | >0.3185 | 0.9677 (0.9397 to 0.9851) | 0.9841 (0.9438 to 0.9981) | 270 | 9 | 124 | 2 | 97.28 | 99.26 | 93.23 |

TP, true positives; TN, true negatives; FP, false positives; FN, false negatives; Accuracy, (TP+TN/TP+FN+TN+FP) ×100; PPV, positive predictive value (TP/TP+FP) ×100; NPV, negative predictive value (TN/TN+FN) ×100.

### 3.5. Results of cross-reactivity

The cross-reactivity of the fusion protein and LPS was assessed by testing them against the panel of sera from SAT-negative patients (n=283) with confirmed active infections by other pathogens (as defined in Methods 2.2). Utilizing iELISA and employing cut-off values established through ROC curve analysis, cross-reactivity with fusion protein was identified in 9 out of 283 serum samples (5 cases of *Escherichia coli* infection, 2 case of *Klebsiella pneumoniae,*1 case of *Pseudomonas aeruginosa,* and 1 case of *Staphylococcus aureus*). Conversely, cross-reactivity with LPS was identified in 41 out of 283 serum samples. The pathogens exhibiting cross-reactivity with LPS included 18 cases of *Escherichia coli* infection, 7 cases of *Klebsiella pneumoniae*, 4 cases of *Pseudomonas aeruginosa*, 3 cases of *Enterococcus faecium,* 2 cases each for *Acinetobacter baumannii, Klebsiella oxytoca, Staphylococcus aureus* and *Streptococcus agalactiae*, and one case of *Serratia marcescens*. The detailed results are presented in S2 File.

## 4. Discussion

Among the currently recognized species of *Brucella*, the six classical species are the primarily cause of human infections, which are predominantly associated with smooth *Brucella* species, such as *Brucella abortus* (*B. abortus*) and *Brucella melitensis* (*B. melitensis*) [24]. The key characteristic of smooth strains is the presence of a complete lipopolysaccharide (LPS) in their outer membrane, featuring an O-polysaccharide chain (O-antigen). This O-antigen is the primary target of the host's antibody response and is thus the basis for most conventional serological tests. In contrast, infections caused by rough *Brucella* species, such as *Brucella canis* (*B. canis*), which lack the O-antigen due to genetic mutations, are relatively infrequent and are notoriously difficult to diagnose with standard O-antigen-dependent assays [25]. Therefore, serological diagnostic methods that utilize LPS as an antigen are inherently limited; they may fail to detect rough *Brucella* infections and are prone to cross-reactivity with other Gram-negative bacteria (e.g., *Yersinia enterocolitica O9*) that share similar O-antigen structures [6,7]. Hence, the identification of proteins that are present in both smooth and rough *Brucella* species may offer a viable solution to this diagnostic challenge [26]. Additionally, the ubiquitous presence of LPS in Gram-negative bacteria heightens the risk of misdiagnosis.

A considerable number of antigenic proteins have been utilized in serological diagnostic studies of brucellosis, including outer membrane proteins; however, the diagnostic efficacy of individual proteins is frequently suboptimal [27]. The fusion of multiple proteins appears to enhance the performance of the assay; nevertheless, the resultant molecular weight is frequently too large for effective recombinant expression. Therefore, early studies have employed bioinformatics tools to predict B-cell epitopes within outer membrane proteins for the tandem construction of multiepitope fusion proteins [10,28]. Although some preliminary results have been obtained, the construction of these fusion proteins has largely been serendipitous due to the inherent uncertainty associated with the predictions generated by bioinformatics tools, indicating significant potential for improvement [11,29–31]. In the present study, we selected B-cell epitopes cataloged in the IEDB for the tandem construction of multi-epitope fusion proteins, which were subsequently experimentally validated, thereby addressing the limitations associated with inaccurate bioinformatics predictions. The present study advances prior foundational research by introducing several novel elements that differentiate it from earlier work. First, experimentally validated B-cell epitopes obtained from the IEDB were employed to construct the fusion protein, thereby avoiding the uncertainties inherent in bioinformatics-based predictions and enhancing the reliability of epitope selection. Second, the fusion protein developed herein demonstrated superior diagnostic performance, achieving an AUC of 0.9912, sensitivity of 0.9534, and specificity of 0.9365, surpassing the metrics reported in previous studies. However, the LPS-based iELISA demonstrated marginally superior metrics. Furthermore, the fusion protein exhibited substantially reduced cross-reactivity, with only 9 of 283 non-*Brucella*-infected samples showing cross-reactivity, in contrast to 41 samples observed with LPS. This reduction is critical for minimizing false-positive results and improving diagnostic specificity. Additionally, the validation process was expanded to include a larger and more diverse cohort, encompassing 283 non-*Brucella*-infected samples, thereby providing more robust evidence for the diagnostic potential of the fusion protein and highlighting its advantages over conventional methods.

The most significant impact of our multiepitope fusion protein on the serological detection of brucellosis lies in its substantial reduction of false-positive results when discriminating brucellosis from other confirmed bacterial infections, a context where LPS-based tests are prone to cross-reactivity, particularly with Gram-negative bacteria. For clinicians and patients, a reduction in false positives means fewer individuals undergoing unnecessary and potentially lengthy antibiotic treatments, reducing associated side effects, healthcare costs, and patient anxiety. Furthermore, it prevents the misallocation of public health resources and allows for a more accurate investigation of true brucellosis outbreaks. However, we should note a limitation of our cross-reactivity assessment. All the cross-reactive sera were confirmed bacterial infection samples. In reality, patients with fever who require differentiation from brucellosis do not have exclusively bacterial infections; other causes include viral and fungal infections. While we tested the fusion protein against a panel of sera from patients with other common infections, the definitive serological status of these samples regarding their respective

pathogens was not reconfirmed in our study. Future studies would benefit from using well-characterized serum panels with confirmed, high-titre antibodies against specific cross-reactive agents (e.g., *Yersinia enterocolitica* O9), providing an even more stringent evaluation of diagnostic specificity.

The 23 epitopes derived from six antigenic proteins within the selected IEDB collection were validated using monoclonal antibodies, these proteins include Cu-Zn SOD, Omp31, DnaK, Omp2b, BP26 and EF Tu [32–44], and the epitopes selected from the IEDB were chosen based on their reported immunogenicity and potential applicability in diagnostic contexts. On the other hand, it is important to note that the IEDB does not provide information regarding the cross-reactivity of these epitopes with other pathogens, a factor that is essential for ensuring the specificity of the diagnostic tool. While these epitopes are highly conserved within *Brucella* species, as indicated by BLAST analysis, certain epitopes are also found to be prevalent in other pathogens. For example, the epitope 'QTREHIL' exhibits 100% similarity with epitopes identified in *Pseudomonas*. Moreover, additional epitopes demonstrate a relatively high degree of homology with those present in *Ochrobactrum*. Following our integration and optimization efforts, we developed a fusion protein that incorporates 23 linear B-cell epitopes, as cataloged by the IEDB. The results of this study confirmed that the protein demonstrated effective performance in the diagnosis of brucellosis, with detection results surpassing those obtained using LPS. In contrast, our serum was classified solely based on serological method (SAT), the absence of culture or PCR confirmation is a limitation of our study, as these methods could provide additional validation of the true status of the serum samples. Future studies should consider incorporating these techniques to enhance the accuracy and reliability of the diagnostic results. While the fusion protein demonstrated high sensitivity (95.34%) in recognizing positive samples, its principal advantage over LPS lies in the significant reduction of cross-reactivity, as detailed in the results.

However, it is likely that the protein's lack of purity contributed to its reduced effectiveness in identifying negative samples. The apparent MW of the purified fusion protein on SDS-PAGE was approximately 45 kDa, which is higher than the theoretical molecular weight of 35.1 kDa predicted from its amino acid sequence. This discrepancy is a common observation with recombinant proteins and can be attributed to several factors: Amino acid composition, the theoretical calculation assumes a uniform peptide-mass ratio, but unusual amino acid composition can affect how the protein binds SDS, altering its electrophoretic mobility. Physicochemical properties: the relatively acidic theoretical pI (4.57) of our fusion protein may influence its SDS-binding capacity and thus its migration. Protein structure: Although SDS-PAGE is run under denaturing conditions, some residual protein structure or the presence of the flexible 'GGGS' linkers might prevent the protein from being fully linearized, leading to anomalous migration. Post-translational modifications: While less common in prokaryotic systems, minor modifications or the presence of the N-terminal methionine can contribute to small shifts in apparent molecular weight. Regarding implementation and cost, the recombinant multiepitope protein offers practical advantages. The iELISA protocol we developed is identical to standard ELISA procedures, requiring no specialized equipment or training, which facilitates its seamless adoption in clinical and public health laboratories. While a precise cost-benefit analysis is beyond the scope of this study, the production of recombinant protein in *E. coli* is generally scalable, reproducible, and cost-effective. In contrast, the extraction and purification of native LPS from pathogenic *Brucella* strains require high-level biosafety containment (BSL-3), which is complex, hazardous, and expensive. Therefore, our recombinant antigen not only eliminates the biosafety risks but is also likely to be more economical and sustainable for large-scale production.

The specific clinical implications of our findings are twofold. Primarily, this fusion protein serves as a superior candidate antigen for developing next-generation commercial ELISA kits. However, whether it can serve as a safer and more specific alternative to LPS requires further validation through the collection of real-world samples. This is particularly crucial in regions where cross-reactive pathogens like *Yersinia enterocolitica O9* are co-endemic. Secondly, its high specificity makes it an excellent confirmatory test following a sensitive screening test, or as a primary diagnostic tool in low-prevalence areas and reference laboratories, where the cost of a false positive is particularly high.

In addition, the results of the cross-reactivity assessment indicated that the fusion protein exhibited cross-reactivity with 9 out of 283 serum samples from bacterial infections. In contrast, LPS showed cross-reactivity with 41 samples,

predominantly associated with *Escherichia coli* infections, which may be attributed to the widespread presence of LPS in Gram-negative bacteria [6,7]. It is crucial to acknowledge that the lack of validation concerning the specificity and coverage of the selected epitopes represents a limitation of our study. While the epitopes were sourced from the IEDB, which contains experimentally validated data, we did not perform individual verification of each epitope's reactivity and coverage across various *Brucella* species. Furthermore, the *Brucella* species infecting the serum donors was not determined, which precluded an assessment of the fusion protein's ability to distinguish between species such as *B. melitensis* and *B. abortus*. Validating each epitope would require additional experiments, including the synthesis of peptides and their conjugation to carriers; processes that are both complex and financially burdensome, exceeding our current funding capabilities. Future research involving larger sample sizes and a more diverse representation of *Brucella* species is essential to thoroughly assess the diagnostic potential of the developed fusion protein.

The multiepitope sequence developed here provides a robust foundation that can be further refined in future iterations. Firstly, the current construct is limited to linear B-cell epitopes expressible in a prokaryotic system. Incorporating conformational epitopes or T-cell epitopes, potentially using eukaryotic expression systems, could enhance its immunogenicity and diagnostic breadth. Secondly, as our epitopes are highly conserved in classical *Brucella* species, this construct is a powerful broad-spectrum tool. However, for species-specific diagnosis (e.g., to distinguish between *B. melitensis* and *B. abortus*), future designs could integrate epitopes unique to specific species. Finally, computational tools for predicting and minimizing homology with human proteins and other common pathogens could be employed to further reduce the risk of autoimmunity or residual cross-reactivity.

Our experiments confirmed the successful construction of a multicellular epitope fusion protein, which exhibits considerable potential for practical applications. A fundamental difference between this study and our previous research resides in the epitope selection methodology. Whereas earlier diagnostic fusion proteins were designed based on epitopes predicted through bioinformatics tools, the present construct is exclusively composed of experimentally validated linear B-cell epitopes obtained from the IEDB. This strategy reduces the uncertainty associated with computational predictions and is grounded in established empirical evidence. Nevertheless, several limitations were identified in our study. The clinical validation of our fusion protein, while promising, was conducted on a cohort where the infecting *Brucella* species was not identified. Future studies involving larger, well-characterized sample sets, including infections caused by rough *Brucella* species such as *B. canis*, will be essential to fully delineate the diagnostic scope and versatility of this assay. Also, it remains unclear whether the patients included in our study were experiencing acute or chronic infections, making it impossible to ascertain whether the protein facilitates the early diagnosis of brucellosis. Future research should prioritize increasing the sample size and utilizing random sampling methods to validate the application potential of the fusion protein.

## 5. Conclusions

In this study, we successfully developed a novel multiepitope fusion protein for the serodiagnosis of brucellosis by integrating experimentally validated linear B-cell epitopes from the IEDB. The fusion protein demonstrated excellent diagnostic performance in iELISA, rivaling that of conventional LPS-based assays. Its most significant advancement lies in a substantial reduction in cross-reactivity with bacteremia infections, effectively addressing a critical flaw of current standard methods. By providing a safer, more specific, and scalable antigen alternative to LPS, this fusion protein represents a promising candidate for next-generation brucellosis diagnostics. Future validation with larger, well-characterized cohorts will be essential to fully translate this robust diagnostic potential into clinical practice.

## Supporting information

**S1 File. Epitope table export results from IEDB.**
(XLSX)

**S2 File. Raw data for I-ELISA.**
(XLSX)

**S3 File. Original images for Fig 2 and I-ELISA results of human serum samples (Integrate 126 negative serum and 283 cross-reactive samples).**
(DOCX)

**S4 File. Blast results for each epitope.**
(XLSX)

## Acknowledgments

We thank the China Animal Health and Epidemiology Center for the gift of LPS and *Brucella* strains, and the Xuzhou Center for Disease Control and Prevention for the gift of human brucellosis sera.

## Author contributions

**Conceptualization:** Dehui Yin.

**Data curation:** Yujia Xie, Shuangshuang Li, Wei Han.

**Funding acquisition:** Yujia Xie, Dehui Yin.

**Methodology:** Tiansong Zhan, Yan Li, Yujia Xie.

**Resources:** Lili Lian.

**Writing – original draft:** Tiansong Zhan, Yan Li, Dehui Yin.

**Writing – review & editing:** Lili Lian, Wei Han, Dehui Yin.

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
