## [Decision Letter · Decision Letter 0]

21 Oct 2025

Multiepitope Fusion Protein-Based ELISA for Enhanced Brucellosis Serodiagnosis

Dear Dr. Yin,

Thank you for submitting your manuscript to PLOS Neglected Tropical Diseases. After careful consideration, we feel that it has merit but does not fully meet PLOS Neglected Tropical Diseases's publication criteria as it currently stands. Therefore, we invite you to submit a revised version of the manuscript that addresses the points raised during the review process.

Please submit your revised manuscript within 60 days Dec 20 2025 11:59PM. If you will need more time than this to complete your revisions, please reply to this message or contact the journal office at plosntds@plos.org. Please include the following items when submitting your revised manuscript:

We look forward to receiving your revised manuscript.

Kind regards,

Richard A. Bowen, DVM PhD

Academic Editor

Elsio Wunder Jr

Section Editor

Shaden Kamhawi

co-Editor-in-Chief

Paul Brindley

co-Editor-in-Chief

**Additional Editor Comments (if provided):**

Your manuscript has been reviewed by 3 experts and each has offered comments to improve your work. Please evaluate and respond to these critiques, edit your manuscript accordingly and re-submit for another round of evakluation. Thank you.

**Journal Requirements:**

**Reviewers' Comments:**

Reviewer's Responses to Questions

**Key Review Criteria Required for Acceptance?**

**Methods**

-Are the objectives of the study clearly articulated with a clear testable hypothesis stated?

-Is the study design appropriate to address the stated objectives?

-Is the population clearly described and appropriate for the hypothesis being tested?

-Is the sample size sufficient to ensure adequate power to address the hypothesis being tested?

-Were correct statistical analysis used to support conclusions?

-Are there concerns about ethical or regulatory requirements being met?

Reviewer #1: See comments below

Reviewer #2: The objectives are clearly stated and clearly addressed. Overall, the methods are thorough, but at times a bit verbose on accepted routine protocols (e.g. transformation). A few brief comments on the construction of the expression vectors would be helpful.

Sample size is appropriate and was clearly addressed from previous comments. Inclusion of positive controls, negative controls and samples from patients with other microbial infections was thorough.

Reviewer #3: - Lines 128-129 - It is not clear how the presence of "varoius pathogens" was determined. The evaluation of cross-reactivity should be performed on samples that contains probable cross-reactive antigens, while here seems that the selection was made only on general diagnosis without testing sera for other pathogens antigens. The absence of a interferent agent don't allow to exclude cross-reactivity.

- The sections 2.5.1 and 2.5.3 are in present tence, like a copy-paste from the experimental procedure: please modify to make in the same form of 2.5.2.

- Lines 249-250 - "who were not diagnosed": the description is incorrect and incomplete. The samples for cross-reactivity must be from individuals for which brucellosis was excluded and that risulted positive for other pathogens or negative at all. In the present form, it sound like the subjects had a possible infection with Brucella not confirmed in laboratory.

**Results**

-Does the analysis presented match the analysis plan?

-Are the results clearly and completely presented?

-Are the figures (Tables, Images) of sufficient quality for clarity?

Reviewer #1: See comments below

Reviewer #2: Figure 1B is of questionable resolution to be informative, but all other figures and tables are clear and informative.

Reviewer #3: - Line 274 - This classifies the fusion protein as stable: why? The reader could not know how to interpret the index: what is the limit vlue to consider a molecule "stable"?

- Sections 3.3 and 3.4 - These two paragraphs are very limited in information. Consider to add more details or to remove 3.3 and 3.4, that are currently useless. It looks like data were already present in existing authors' works reported in references.

- Section 3.5 - How the reactivity against LPS by other pathogens was evaluated? As reported in Methods comments, it is necessary to confirm the presence of the antigen produced by other pathogens in the samples, otherwise a negative result could be attributed to the absence of the interfering substrate.

**Conclusions**

-Are the conclusions supported by the data presented?

-Are the limitations of analysis clearly described?

-Do the authors discuss how these data can be helpful to advance our understanding of the topic under study?

-Is public health relevance addressed?

Reviewer #1: See comments belos

Reviewer #2: How does this change the outcome of serological detection of Brucella? How easily could this be implemented and what is the cost difference between LPS and purified recombinant multiepitope protein? How could this multiepitope sequence be further improved? What are the specific clinical implications? What is the significance of reduced false positives?

The conclusions are supported by the data and analysis. The limitations are sometimes too explicit (lack of Brucella information,

Reviewer #3: - Lines 364-370 and 415 - There ae many discrepancies with data in other sections, that must be clarified. At present, these results do not include all samples collected and partially describe and support the study outcomes.

- Lines 411-412 - "the fusion protein demonstrated comparable efficacy in recognizing positive samples when compared to LPS." Data on this are not presented in the work.

**Editorial and Data Presentation Modifications?**

Reviewer #1: See below

Reviewer #2: Line 39 Incomplete sentence.

Line 39, Line 58, Line 260, Line 265, Line 403, Table 1, conflicting statements of 23 epitope construct and 11 epitopes

Line 279 The <predicted> 3D structure of the protein is illustrated in Figure 1.

Line 428 change multiple hyphens to semicolon.

Line 440 typographical error "Aslo"

Clarify what grayscale analysis means. What method was used to determine purity of the final protein?

Explanation of how the identity of the final protein was determined, possible reasons the observed MW of the protein is significantly different than the predicted size.

The manuscript could benefit from more explicit statements on the larger impact and significance of the research.

Was endotoxin (LPS contamination) measured in the final purified protein sample?</predicted>

Reviewer #3: - Line 39 - Should the verb be "comprised"?

- Line 77 - Would it be possible to replace reference 2 with a more recent one? Perhaps using 3 also here.

- Line 83 - Cross-reaction is detected for many other pathogens, while here it seems that is limited to only these two. Please change (for example: cross-reactivity with other bacteria such as Yersinia ...).

- Line 84 - "contemporary research" has an unscientific soundness: please try to modify.

- Line 87 - "in contemporary research" is a repetition of previous sentence: please modify.

- Lines 89-91 - Make the sentence more linear and clear.

- Line 91 - "In this methodology,": remove and add and adverb in the sentence (for example: ...proteins is therefore of...).

- Line 94 - The work in reference 12 did not conclude the superior performance of LPS.

- Lines 99-101 - A reference is needed.

- Line 129 - "which collected": a verb is missing.

- Line 265 - "Refer to" in brackets is not necessary.

- Lines 266-268 - It would be better to change "are particularly" in something like "was found to be conserved among most relevant species..."

- Line 304 - Is it "Cause of human"?

- Lines 423-424 - It is impossible to deduce species from sera with the available information: it would be better to explain that the specie infecting serum donors was not determine.

- Lines 434-436 - Please put in evidence what is the diffrence with previous works of the authors.

**Summary and General Comments**

Reviewer #1: Abstract: the sentence ‘lower specificity, but less cross-reactivity’ seems a paradox

Author’s summary:

Line 64-65: the numbers in these sentences contradict the results in the abstract

Line 83: Yersinia enterocolitica O9 and Escherichia coli O157:H7: where these included in controls?

Line 125: had positive controls also positive blood cultures? If so, which Brucella species were found? If not, how certain are the authors that these were true Brucella samples? Was there seroconversion? A single positive serology test does not provide an accurate diagnosis of brucellosis.

Table 2: the abstract says ‘9’ cross reactivity in the fusion protein group, but there are only ‘8’ FPs; there are ‘39’ that showed cross reactivity in the LPS group, but only ‘2’ FP in the table

It is confusing that in the methods section, line 126 describes healthy individuals as controls, and on line 128, samples from febrile patients with pathogens are used to check for cross-reactivity. It should be noted that the latter group is, in the real world, a much more realistic and effective control group than the first group. One can question why the Table 2 and ROC curve were not created with samples from febrile patients with pathogens as the control.

In the results section, tables, and figures, it is not clearly stated which of the two control groups was used, which is critical for correct interpretation. Also, in the abstract, it is not possible to know that there are actually two different control groups, and the results section mixes up both.

Discussion:

Explain the role of the O-antigen in the rough vs smooth Brucella species

The discussion is lengthy, and many paragraphs discuss issues related to the specific antigens, which are less relevant to the discussion. The core statistical findings, along with their strengths, limitations, and implications for clinical practice, are not discussed.

Conclusion:

Limitations raised here are part of the discussion, not the conclusion. The findings should be summarized as well as their importance.

Reviewer #2: The strongest section is the discussion section, and the manuscript could benefit from expanded analysis of other sections.

Reviewer #3: The work was well written and understandable. However, crucial data are missing, in particular regarding testing for cross-reactivity. The authors have other works with similar topics, but the data therein presented and of interest for the present paper must be included: a reader must have all data needed in the single work.

PLOS authors have the option to publish the peer review history of their article (what does this mean? ). If published, this will include your full peer review and any attached files.

**Do you want your identity to be public for this peer review?** For information about this choice, including consent withdrawal, please see our Privacy Policy .

Reviewer #1: **Yes: ** Steven Van Den Broucke

Reviewer #2: No

Reviewer #3: No

**Figure resubmission:**
---

## [Decision Letter · Decision Letter 1]

21 Nov 2025

Multiepitope Fusion Protein-Based ELISA for Enhanced Brucellosis Serodiagnosis

Dear Dr. Yin,

Thank you for submitting your manuscript to PLOS Neglected Tropical Diseases. After careful consideration, we feel that it has merit but does not fully meet PLOS Neglected Tropical Diseases's publication criteria as it currently stands. Therefore, we invite you to submit a revised version of the manuscript that addresses the points raised during the review process.

Please submit your revised manuscript within by Dec 21 2025 11:59PM. If you will need more time than this to complete your revisions, please reply to this message or contact the journal office at plosntds@plos.org. Please include the following items when submitting your revised manuscript:

We look forward to receiving your revised manuscript.

Kind regards,

Richard A. Bowen, DVM PhD

Academic Editor

Elsio Wunder Jr

Section Editor

Shaden Kamhawi

co-Editor-in-Chief

Paul Brindley

co-Editor-in-Chief

**Additional Editor Comments (if provided):**

Your manuscript has been reviewed by 3 experts and collectively they have expressed a number of concerns and provided several suggestions that you need to address. Please review these comments carefully, modify you manuscript accordingly and provide a "response to reviewers" document indicating your responses to these reviews. Thank you.

**Journal Requirements:**

**Reviewers' Comments:**

Reviewer's Responses to Questions

**Key Review Criteria Required for Acceptance?**

**Methods**

-Are the objectives of the study clearly articulated with a clear testable hypothesis stated?

-Is the study design appropriate to address the stated objectives?

-Is the population clearly described and appropriate for the hypothesis being tested?

-Is the sample size sufficient to ensure adequate power to address the hypothesis being tested?

-Were correct statistical analysis used to support conclusions?

-Are there concerns about ethical or regulatory requirements being met?

Reviewer #1: (No Response)

**Results**

-Does the analysis presented match the analysis plan?

-Are the results clearly and completely presented?

-Are the figures (Tables, Images) of sufficient quality for clarity?

Reviewer #1: (No Response)

**Conclusions**

-Are the conclusions supported by the data presented?

-Are the limitations of analysis clearly described?

-Do the authors discuss how these data can be helpful to advance our understanding of the topic under study?

-Is public health relevance addressed?

Reviewer #1: (No Response)

**Editorial and Data Presentation Modifications?**

Reviewer #1: (No Response)

**Summary and General Comments**

Reviewer #1: Line 369: to be clarified that fewer false positives were in the comparison with confirmed bacterial infections, which will have more Gram-negative bacteria that cross-react with LPS. In the ‘febrile’ control group, which might better reflect the real world, there were no fewer false positives.

Line 433: same comment as above. Real-world febrile patients are not synonymous with confirmed bacterial positive samples. In the ROC curve, representing real-world ‘febrile’ patients, LPS outperformed the fusion protein

Line 490: cross-reactivity with bacteremic infections (cfr above)

PLOS authors have the option to publish the peer review history of their article (what does this mean? ). If published, this will include your full peer review and any attached files.

**Do you want your identity to be public for this peer review?** For information about this choice, including consent withdrawal, please see our Privacy Policy .

Reviewer #1: **Yes: ** Steven Van Den Broucke

**Figure resubmission:**
---

## [Editor Report · Decision Letter 2]

28 Nov 2025

Dear Dr Yin,

We are pleased to inform you that your manuscript 'Multiepitope Fusion Protein-Based ELISA for Enhanced Brucellosis Serodiagnosis' has been provisionally accepted for publication in PLOS Neglected Tropical Diseases.

Best regards,

Richard A. Bowen, DVM PhD

Academic Editor

Elsio Wunder Jr

Section Editor

Shaden Kamhawi

co-Editor-in-Chief

Paul Brindley

co-Editor-in-Chief

Thank you for your patience in our review process and for addressing issues raised by reviewers.

---

## [Editor Report · Acceptance letter]

Dear Dr Yin,

We are delighted to inform you that your manuscript, "Multiepitope Fusion Protein-Based ELISA for Enhanced Brucellosis Serodiagnosis," has been formally accepted for publication in PLOS Neglected Tropical Diseases.

Best regards,

Shaden Kamhawi

co-Editor-in-Chief

Paul Brindley

co-Editor-in-Chief
